# Transfer accuracy of partially enclosed single hard vacuum-formed trays with 3D-printed models for lingual bracket indirect bonding: A prospective in-vivo study

**Viet Anh Nguyen**[1]*, **Thuy Anh Nguyen**[2], **Hong Le Doan**[2], **Thi Hanh Pham**[2], **Bich Ngoc Doan**[2], **Thi Thanh Thuy Pham**[2], **Viet Hoang**[3]

**1** Faculty of Dentistry, PHENIKAA University, Hanoi, Vietnam, **2** Private Practice, Viet Anh Orthodontic Clinic, Hanoi, Vietnam, **3** Faculty of Dentistry, Van Lang University, Ho Chi Minh, Vietnam

* anh.nguyenviet1@phenikaa-uni.edu.vn

**Data Availability Statement:** All relevant data are within the manuscript and its Supporting Information files.

## Abstract

### Objective

This study aims to evaluate the clinical transfer accuracy of partially enclosed single hard vacuum-formed trays based on three-dimensional (3D) printed models for lingual bracket indirect bonding.

### Materials and methods

Thirty-two consecutive patients receiving lingual orthodontic treatment were enrolled. Digital models with ideal bracket positions were 3D-printed, followed by fabrication of partially enclosed single hard vacuum-formed trays. Digital impressions captured actual bracket positions and were compared to the ideal positions. One-tailed t-tests assessed if errors were within clinically acceptable thresholds of 0.5 mm for linear measurements and 2° for angular measurements.

### Results

Mean bracket transfer errors were 0.052 mm, 0.076 mm, 0.106 mm, 0.795°, 1.344°, and 2.485° for mesiodistal, buccolingual, occlusogingival, rotation, tip, and torque, respectively. Transfer errors were statistically below the clinically acceptable thresholds for all dimensions except torque. Frequencies of acceptable transfer errors were 100%, 100%, 99.3%, 93.1%, 78.3%, and 54.0%, respectively.

### Conclusion

Partially enclosed single hard vacuum-formed trays with 3D-printed models transfer lingual brackets with high accuracy in the mesiodistal, buccolingual, and occlusogingival dimensions, rotation, and tip. However, the transfer of torque remains questionable.

**Funding:** The author(s) received no specific funding for this work.

## Introduction

Bracket transfer accuracy is crucial in lingual orthodontic treatment, securing the precise transfer of prescription and overcorrection from the ideal setup to the patient's dentition [1]. Digital workflows with three-dimensional (3D) printing technology may simplify the manufacturing process of indirect bonding trays [2,3]. Additionally, the transfer accuracy may be enhanced compared to analog ones due to the ability to transfer a group of teeth simultaneously with greater anatomical guidance, achieved by virtual repositioning of teeth and accompanying brackets to the initial malocclusion state [4].

Double vacuum-formed indirect bonding trays are widely used and demonstrated to accurately transfer both labial and lingual brackets in linear and rotational dimensions [5,6]. However, the transfer accuracy of tip and torque remains questionable. This inaccuracy may result from the flexibility of the inner soft layer and the possible separation between the two layers. Additionally, double vacuum-formed trays cannot be used for single-tooth bonding, in cases of severely crowded teeth or bracket failures, due to the lack of a passive fit between trays and teeth [2]. Therefore, a modified design of vacuum-formed trays with a single hard layer is proposed.

This study aims to evaluate the clinical transfer accuracy of partially enclosed single hard vacuum-formed trays based on 3D-printed models for lingual bracket indirect bonding. The null hypothesis stated that bracket transfer errors were non-inferior to the clinically acceptable thresholds of 0.5 mm and 2˚.

## Materials and methods

### Subject

This prospective clinical study was approved by the Institutional Ethical Review Board at Hanoi Medical University (protocol no. 2301). All participants were thoroughly counseled about the study and signed informed consent. A sample of 32 patients, including 3 males and 29 females, aged 21–35 years, were consecutively enrolled from September 9[th] 2023 to March 7[th] 2024. Of these, 13 patients received lingual appliances on both arches, while 19 received them only on the upper arch. The arch-length discrepancies of the upper and lower arches were 3.42 ± 2.85 mm and 6.89 ± 3.30 mm, respectively. Non-extraction treatment was provided for 12 patients, while premolar extraction treatment was performed on 20 patients.

A sample size calculation, based on a previous study's effect size of 0.149 for vacuum-formed trays in lingual bracket indirect bonding, determined that 279 brackets were needed to achieve 80% power in detecting statistically significant mean transfer errors below 0.5 mm or 2˚ using one-sample t-tests at a significance level of 0.05 [6].

### Tray fabrication and lingual bracket bonding

Digital impressions were taken with an i700 intraoral scanner (Medit, Seoul, Korea). The scan data were imported into Autolign orthodontic software (Diorco, Gyeonggi-do, Korea) for tooth segmentation and ideal setup creation. ADB lingual brackets (Medico, Gyeonggi-do, Korea) were positioned with the straight archwire concept on the ideal setupand virtually moved together with corresponding teeth back to the initial malocclusion state [7]. The gaps between the brackets and teeth were virtually filled. Additionally, bracket slots and undercuts below bracket wings were virtually blocked out to facilitate subsequent bracket placement and tray removal. The resulting virtual models with ideal bracket positions were 3D-printed with a Photon D2 digital light processing printer (Anycubic, Shenzen, China).

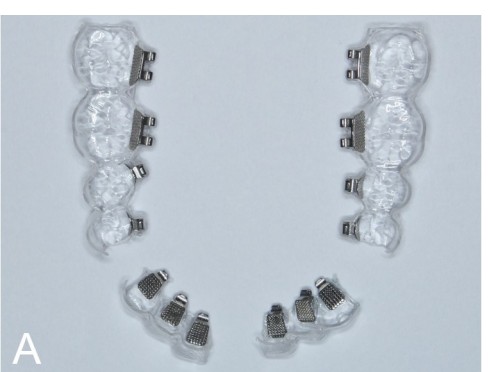
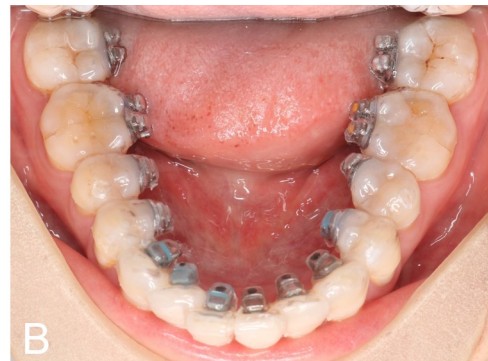

**Fig 1. Partially enclosed single vacuum-formed indirect bonding trays.** (A) Segmented trays with lingual brackets inserted. (B) Tray placement on clinical bonding.

Indirect bonding trays were vacuum-formed on the 3D-printed models with single Biocryl hard foils of 1 mm thickness (Scheu Dental, Iserlohn, Germany). After being removed from the models, the trays were sectioned into four parts, including two anterior and two posterior segments. The gingival wall of all bracket lodgements was removed with a rotary disc to create a partially enclosed design, facilitating excess adhesive and tray removal clinically (Fig 1).

On clinical bonding, the lingual tooth surfaces were pumiced and etched with 37% phosphoric acid for 20 seconds, followed by rinsing and drying. Assure Plus primer (Reliance, Itasca, IL) was applied to the tooth surfaces and GoTo adhesive (Reliance) was applied to the bracket bases. The tray was then seated with light finger pressure on the occlusal surface, followed by excess flash removal and light-curing for 40 seconds. Pointed burs were used to partially grind out hard tray materials around the brackets, facilitating tray removal without debonding the brackets.

## Data acquisition

Post-bonding digital impressions were taken with the same intraoral scanner to assess the discrepancy between actual and planned bracket positions. The measurement of bracket transfer error followed a previously established methodology [6]. The post-bonding scan data and the virtual model with ideal bracket positions, serving as the target and reference data, were first roughly aligned. Then, they were segmented into individual teeth, removing the gingiva and retaining only the clinical crowns and brackets. This initial alignment ensured consistent cropping and facilitated the subsequent superimposition based on the tooth surface. The centers of the manufacturer-provided virtual bracket patches without bracket bases were set at the origin of a coordinate system, with the x, y, and z-axis parallel to the mesiodistal edge, vertical edge, and buccolingual edge of the bracket slot, respectively. For each tooth, the brackets on both the target and reference data were initially aligned with the bracket patch, constituting the first and second superimpositions. Subsequently, the bracket patch was attached to the target data followed by superimposing the target data and bracket patch combination onto the reference data using a local best-fit algorithm that considered only the tooth surfaces. The new coordinates of the bracket patch's center in the x, y, and z directions would indicate the bracket linear transfer errors. Meanwhile, the angular transfer errors would be represented by the rotation of the bracket slot's edges projected onto the respective coordinate planes (Fig 2). The 3D inspection software (Meshmixer, Autodesk, USA) was utilized to obtain these measurements.

For each bracket, linear transfer errors were presented in the mesiodistal, buccolingual, and occlusogingival dimensions. Additionally, angular transfer errors were described as rotation,

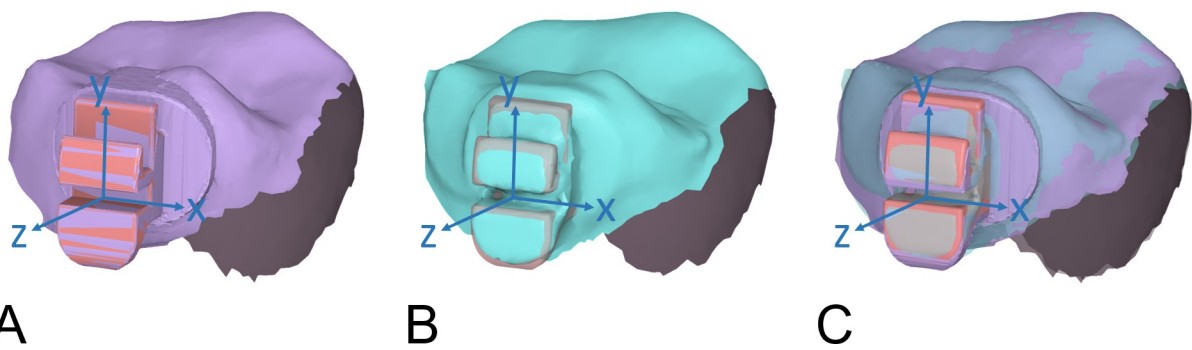

**Fig 2. Methodology for measuring bracket transfer errors.** (A) Align the reference data with the bracket patch set at the coordinate origin. (B) Align the target data with the bracket patch set at the coordinate origin. (C) Superimpose the target data with the reference data to calculate transfer errors.

tip, and torque. Absolute values were calculated for each component to prevent the cancellation between positive and negative values. To assess measurement reliability, a second examiner remeasured 50 brackets followed by calculating the intraclass correlation coefficient (ICC).

## Statistical analysis

Data analyses were performed using SPSS 23.0 software (IBM, Armonk, NY) with the statistical significance level set to $\alpha = 0.05$. The data were assessed for the normal distribution using Kolmogorov-Smirnov tests. Transfer errors of each tooth group and the entire sample were presented as means and standard deviations. Two-sample t-tests comparing right and left values revealed no significant differences, warranting the combination of right and left teeth in the same tooth group for further analysis.

One-tailed t-tests were performed to determine whether the bracket transfer errors were statistically within the clinically acceptable thresholds of 0.5 mm and 2° for linear and angular dimensions, respectively. These clinically acceptable thresholds, widely used in previous studies on bracket transfer accuracy, were employed based on the objective grading system of the American Board of Orthodontics [4,5,8]. Additionally, the percentages of bracket transfer errors within the clinically acceptable thresholds were described.

## Results

A single clinician bonded 559 lingual brackets, of which 17 debonded during tray removal, resulting in a 3.04% debonding rate. Due to crowding, bracket placement was not possible on 18 teeth during the initial bonding appointment. After excluding 3 brackets due to poor scan quality, 539 brackets were analyzed to assess the transfer accuracy. The total bonding time, from the initial placement of the first tray segment to the removal of the final segment, was 22.23 ± 6.13 minutes for the upper arch and 21.77 ± 5.31 minutes for the lower arch. The inter-examiner ICCs were 0.965 for linear dimensions and 0.944 for angular dimensions, indicating the high reliability of the measuring methodology.

Table 1 presents the means and standard deviations of bracket transfer errors and the result of one-tailed t-tests. Fig 3 shows box-and-whisker plots of bracket transfer errors for each tooth group. Of the entire sample, the linear transfer errors were 0.052 ± 0.044 mm, 0.076 ± 0.065 mm, and 0.106 ± 0.098 mm for the mesiodistal, buccolingual, and occlusogingival dimensions, respectively. The angular transfer errors of the entire sample were

**Table 1. Descriptive statistics of bracket transfer errors for each tooth type and the entire sample[a,b].**

| Tooth type | n | Linear dimension | | | Angular dimension | | |
|---|---|---|---|---|---|---|---|
| | | MD (mm) | BL (mm) | OG (mm) | Rotation (°) | Tip (°) | Torque (°) |
| Incisor | 171 | 0.049 ± 0.042[c] | 0.066 ± 0.062[c] | 0.092 ± 0.093[c] | 0.823 ± 0.734[c] | 1.529 ± 1.412[c] | 2.458 ± 2.227 |
| Upper | 125 | 0.051 ± 0.044[c] | 0.065 ± 0.059[c] | 0.088 ± 0.088[c] | 0.786 ± 0.737[c] | 1.507 ± 1.430[c] | 2.593 ± 2.252 |
| Lower | 46 | 0.045 ± 0.036[c] | 0.069 ± 0.072[c] | 0.103 ± 0.105[c] | 0.924 ± 0.727[c] | 1.590 ± 1.375[c] | 2.089 ± 2.138 |
| Canine | 74 | 0.060 ± 0.046[c] | 0.062 ± 0.054[c] | 0.088 ± 0.081[c] | 0.791 ± 0.704[c] | 1.151 ± 0.919[c] | 2.171 ± 2.091 |
| Upper | 53 | 0.060 ± 0.047[c] | 0.059 ± 0.050[c] | 0.090 ± 0.085[c] | 0.723 ± 0.718[c] | 1.075 ± 0.935[c] | 2.211 ± 2.027 |
| Lower | 21 | 0.062 ± 0.042[c] | 0.068 ± 0.064[c] | 0.081 ± 0.073[c] | 0.960 ± 0.652[c] | 1.342 ± 0.868[c] | 2.069 ± 2.295 |
| Premolar | 122 | 0.061 ± 0.052[c] | 0.096 ± 0.071[c] | 0.133 ± 0.122[c] | 0.813 ± 0.743[c] | 1.395 ± 1.145[c] | 2.563 ± 2.562 |
| Upper | 90 | 0.061 ± 0.054[c] | 0.100 ± 0.074[c] | 0.141 ± 0.128[c] | 0.861 ± 0.797[c] | 1.411 ± 1.125[c] | 2.805 ± 2.821 |
| Lower | 32 | 0.060 ± 0.047[c] | 0.082 ± 0.061[c] | 0.111 ± 0.104[c] | 0.678 ± 0.552[c] | 1.352 ± 1.216[c] | 1.885 ± 1.460 |
| Molar | 172 | 0.046 ± 0.039[c] | 0.078 ± 0.064[c] | 0.109 ± 0.086[c] | 0.755 ± 0.701[c] | 1.206 ± 0.855[c] | 2.591 ± 2.324 |
| Upper | 120 | 0.047 ± 0.041[c] | 0.070 ± 0.057[c] | 0.110 ± 0.078[c] | 0.908 ± 0.743[c] | 1.280 ± 0.895[c] | 2.664 ± 2.464 |
| Lower | 52 | 0.046 ± 0.031[c] | 0.096 ± 0.075[c] | 0.106 ± 0.103[c] | 0.404 ± 0.421[c] | 1.038 ± 0.737[c] | 2.423 ± 1.973 |
| Total | 539 | 0.052 ± 0.044[c] | 0.076 ± 0.065[c] | 0.106 ± 0.098[c] | 0.795 ± 0.720[c] | 1.344 ± 1.138[c] | 2.485 ± 2.318 |

[a] Results are presented as means ± standard deviations.

[b] n, number of analyzed brackets in each tooth group; BL, buccolingual; MD, mesiodistal; OG, occlusogingival.

[c] Bracket transfer errors are significantly within the thresholds of 0.5 mm and 2°, as determined by one-sided t-tests.

0.795 ± 0.720°, 1.344 ± 1.138°, and 2.485 ± 2.318° for rotation, tip, and torque, respectively. Comparisons between upper and lower teeth, as well as among different tooth groups, revealed no consistent patterns, with the exception of consistently lower angular transfer errors observed in the lower premolar and molar brackets compared to their upper counterparts.

One-tailed t-tests revealed statistical significance (P < .05) for all linear dimensions across all tooth groups. Regarding angular dimensions, one-tailed t-tests indicated statistical

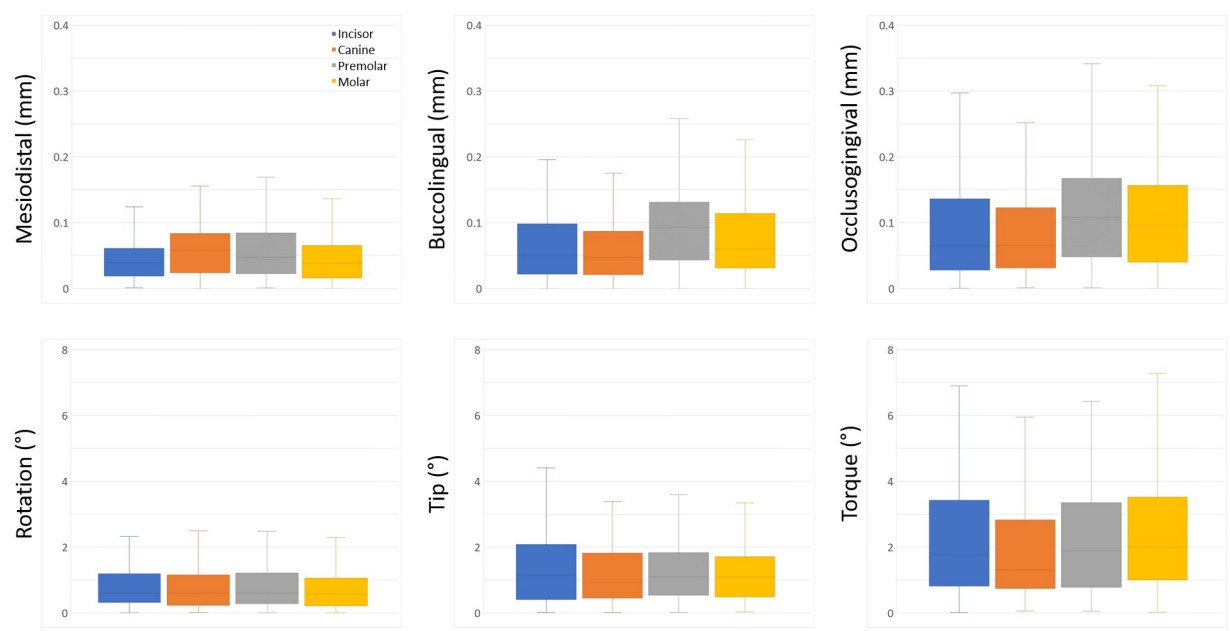

**Fig 3. Box-and-whisker plots of bracket transfer errors for four tooth groups in linear and angular dimensions.**

**Table 2. Frequencies of bracket errors falling within the clinically acceptable thresholds[a,b].**

| Tooth type | n | Linear dimension | | | Angular dimension | | |
|---|---|---|---|---|---|---|---|
| | | Mesiodistal | Buccolingual | Occlusogingival | Rotation | Tip | Torque |
| Incisor | 171 | 171 (100) | 171 (100) | 169 (98.8) | 159 (93.0) | 125 (73.1) | 96 (56.1) |
| Upper | 125 | 125 (100) | 125 (100) | 124 (99.2) | 117 (93.6) | 94 (75.2) | 67 (53.6) |
| Lower | 46 | 46 (100) | 46 (100) | 45 (97.8) | 42 (91.3) | 31 (67.4) | 29 (63.0) |
| Canine | 74 | 74 (100) | 74 (100) | 74 (100) | 68 (91.9) | 61 (82.4) | 45 (60.8) |
| Upper | 53 | 53 (100) | 53 (100) | 53 (100) | 48 (90.6) | 44 (83.0) | 33 (62.3) |
| Lower | 21 | 21 (100) | 21 (100) | 21 (100) | 20 (95.2) | 17 (81.0) | 12 (57.1) |
| Premolar | 122 | 122 (100) | 122 (100) | 121 (99.2) | 116 (95.1) | 96 (78.7) | 64 (52.5) |
| Upper | 90 | 90 (100) | 90 (100) | 89 (98.9) | 84 (93.3) | 71 (78.9) | 44 (48.9) |
| Lower | 32 | 32 (100) | 32 (100) | 32 (100) | 32 (100) | 25 (78.1) | 20 (62.5) |
| Molar | 172 | 172 (100) | 172 (100) | 171 (99.4) | 159 (92.4) | 140 (81.4) | 86 (50.0) |
| Upper | 120 | 120 (100) | 120 (100) | 120 (100) | 107 (89.2) | 95 (79.2) | 59 (49.2) |
| Lower | 52 | 52 (100) | 52 (100) | 51 (98.1) | 52 (100) | 45 (86.5) | 27 (51.9) |
| Total | 539 | 539 (100) | 539 (100) | 535 (99.3) | 502 (93.1) | 422 (78.3) | 291 (54.0) |

[a] n, number of analyzed brackets in each tooth group.

[b] Results are expressed as numbers (percentages).

significance (P < .05) for rotation and tip, but not for torque, in all tooth groups. Within each tooth type, linear transfers generally demonstrated greater accuracy than angular transfers. Among linear transfer errors, the occlusogingival dimension typically showed the highest magnitude, followed by the buccolingual and then mesiodistal dimensions, except for upper canines. For angular dimensions, torque was consistently the least accurate, followed by tip and then rotation across all tooth types.

Table 2 shows the percentages of bracket transfer errors that fall within the clinically acceptable thresholds of 0.5 mm and 2˚ for each tooth group. Regarding linear errors, 100% of brackets exhibited mesiodistal and buccolingual transfer errors within the acceptable threshold, while 99.3% of brackets demonstrated acceptable occlusogingival transfer accuracy. Among angular errors, the percentage of rotational accuracy was highest at 93.1%, while that of torque was lowest at 54.0%. Bracket transfer errors within the acceptable thresholds occur with similar frequency across all tooth types without consistent patterns.

## Discussion

One-tailed t-tests revealed that linear bracket transfer errors for mesiodistal, buccolingual, and occlusogingival dimensions were statistically significantly within the clinically acceptable threshold of 0.5 mm. Similarly, angular transfer errors for rotation and tip were statistically significantly less than the 2˚ threshold, while torque errors did not show this significance level. Thus, the null hypothesis was rejected.

The higher proportion of female patients in this study aligns with findings from previous research, which may reflect differing aesthetic preferences or priorities between genders [4,6,9]. The bracket debonding rate with single hard vacuum-formed trays (3.04%) was higher in this study compared to the rate reported by Anh et al (0.98%) using double vacuum-formed trays [6]. This difference may be attributable to the increased rigidity of the single hard trays, potentially making tray removal more challenging, especially in cases of crowding requiring varied bracket insertion directions. Additionally, bonding time was longer with single hard vacuum-formed trays compared to the double trays used by Anh et al. This could be due to the

need for partial grinding of the tray material around each bracket to facilitate tray removal, a process requiring additional time.

Conversely, the increased rigidity of single hard trays, compared to the inner soft layer of double vacuum-formed trays, may provide greater spatial stability, potentially resulting in improved angular accuracy [6]. This is supported by the finding that angular transfer errors in this study were lower than those reported by Anh et al using double trays. Furthermore, the increased rigidity of single hard trays may facilitate a more passive fit. This is particularly beneficial for single-tooth bonding where anatomical landmarks are limited, a recognized limitation of double vacuum-formed trays. This advantage becomes even more significant with lingual appliances, as the shorter inter-bracket spans in lingual orthodontics may necessitate bonding teeth sequentially rather than all at once at the beginning of treatment. However, transferring a group of teeth simultaneously during initial bonding is preferred to enhance anatomical guidance, as evidenced by the superior bracket transfer accuracy observed in this study compared to a previous study using individual bracket transfer jigs [10]. Single vacuum-formed trays also offer the advantage of material saving compared to double trays [11].

This study assesses a modified design for vacuum-formed trays, diverging from the current trend of utilizing directly 3D-printed trays for indirect bonding. However, using single vacuum-formed trays with 3D-printed models is more cost-effective than direct 3D-printing of trays, due to simplified design and reduced post-processing requirements [12]. As no studies in the literature have evaluated the accuracy of 3D-printed indirect bonding trays for lingual brackets, this study's findings are compared with those from a study using 3D-printed trays for labial brackets [4]. That study reported mean transfer errors of 0.10 mm mesiodistally, 0.10 mm buccolingually, 0.18 mm occlusogingivally, 2.47° in rotation, 2.01° in tip, and 2.55° in torque. While the linear transfer errors were within the clinically acceptable threshold, the angular errors were not. Notably, all transfer errors in that study exceeded those observed in the current study. This discrepancy may be attributed to the flexibility of the tray material and the inadequate bracket retention within the tray, as silicone and modeling wax were necessary to secure brackets in a similar study utilizing 3D-printed trays [9]. Furthermore, as both the 3D-printing material and the orthodontic adhesive are light-cured, their adhesive natures may interact, potentially hindering tray removal and causing tearing [12].

In this study, hard foils with an initial thickness of 1 mm were used, resulting in an approximate 0.5 mm thickness after vacuum forming. Thinner 0.8 mm foils were not suitable, as they failed to securely hold brackets during preliminary testing. Conversely, thicker 1.5 mm foils hindered complete bracket insertion, likely due to the increased rounding of bracket lodgement edges during cooling, leading to a mismatch with the bracket shape. The partially enclosed design of bracket lodgements, similar to that of 3D-printed indirect bonding trays, facilitates both excess flash removal and tray removal [12–15]. This is advantageous compared to fully enclosed designs, where the gingival wall may impede tray removal and increase the risk of bracket debonding, particularly with hard tray materials. Blocking out bracket slots and undercuts is necessary to prevent tray material from entering these spaces, ensuring that brackets can be fully inserted into their lodgements.

Although the lower arch exhibited more crowding than the upper arch, no consistent pattern in transfer accuracy was observed between the upper and lower anterior teeth. This is attributed to the exclusion of severely crowded teeth from the initial bonding and subsequent analysis. The higher angular transfer accuracy of lower premolar and molar brackets compared to their upper counterparts may be attributed to enhanced visibility in the lower arch during lingual bracket bonding, allowing for easier verification of correct bracket and tray positioning.

The higher linear transfer accuracy compared to angular accuracy aligns with previous studies on both lingual and labial brackets, suggesting that angular bracket positioning is inherently less stable than linear positioning [1,4,5,14–16]. Additionally, challenges in determining bracket axes on scan data due to nonparallel edges and distorted surfaces caused by rounding effects, as well as limitations in scanning reflective metal surfaces, may further contribute to angular transfer errors [6].

The mesiodistal dimension exhibited the highest linear transfer accuracy, likely due to the secure hold provided by both the mesial and distal walls of the bracket lodgements. Buccolingual transfer accuracy was slightly lower, potentially because while bracket positioning in this dimension is influenced by both lingual walls and tooth surfaces, the designed gap between the bracket base and tooth surface reduces the tooth surface's stabilizing effect. The lowest linear transfer accuracy was observed in the occlusogingival dimension, likely due to vertical bracket positions being controlled solely by the occlusal walls of the lodgements.

Similarly, for angular accuracy, the highest control was seen in rotation, due to the combined influence of mesial, distal, lingual walls, and tooth surfaces. Tip control was moderate, primarily influenced by the mesial and distal walls. Torque exhibited the lowest angular transfer accuracy, likely due to control primarily from the lingual walls and only partial control from the tooth surfaces.

The high bracket transfer accuracy, with mean transfer errors within clinically acceptable limits in five out of six analyzed dimensions and a nearly 100% rate of accurate linear transfer, demonstrates the compatibility of single hard vacuum-formed trays with 3D-printed models for clinical lingual bracket bonding. Any suboptimal tooth positions resulting from significant transfer errors can be corrected during the finishing stage through wire bending or bracket repositioning.

This study has several limitations. First, it lacked control groups using alternative methods such as double vacuum-formed or directly 3D-printed indirect bonding trays. Second, transfer accuracy for single-tooth bonding was not assessed. Additionally, only one type of lingual bracket was evaluated. Future studies should compare the transfer accuracy of various indirect bonding tray types across different lingual bracket designs.

## Conclusion

Our study showed high lingual bracket transfer accuracy of partially enclosed single hard vacuum-formed trays with 3D-printed models in the mesiodistal, buccolingual, and occlusogingival dimensions, rotation, and tip. However, the transfer of torque remains questionable. Linear transfer accuracy is generally higher than angular transfer accuracy. Bracket transfer errors outside the acceptable thresholds exhibit similar frequency across all tooth types.

## Supporting information

**S1 File. Dataset.**
(XLSX)

**S2 File. Reporting checklist for cohort study.**
(DOCX)

## Author Contributions

**Conceptualization:** Viet Anh Nguyen, Thi Thanh Thuy Pham.

**Data curation:** Viet Anh Nguyen.

**Methodology:** Viet Anh Nguyen, Hong Le Doan.

**Software:** Viet Anh Nguyen, Thi Hanh Pham.

**Supervision:** Viet Anh Nguyen.

**Validation:** Thuy Anh Nguyen.

**Writing – original draft:** Viet Anh Nguyen, Thuy Anh Nguyen, Bich Ngoc Doan.

**Writing – review & editing:** Viet Anh Nguyen, Viet Hoang.

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
