## [Decision Letter · Decision Letter 0]

10 Nov 2024

PONE-D-24-35775Transfer accuracy of partially enclosed single hard vacuum-formed trays with 3D-printed models for lingual bracket indirect bonding: A prospective in-vivo studyPLOS ONE

Dear Dr. Nguyen,

Thank you for submitting your manuscript to PLOS ONE. After careful consideration, we feel that it has merit but does not fully meet PLOS ONE’s publication criteria as it currently stands. Therefore, we invite you to submit a revised version of the manuscript that addresses the points raised during the review process. We found your work to be both relevant to the field and methodologically sound. The study design demonstrates careful consideration, and the manuscript is generally well-written.

However, some points need further description in the text or clarification, as pointed by the reviewers.

 Please submit your revised manuscript by Dec 25 2024 11:59PM. If you will need more time than this to complete your revisions, please reply to this message or contact the journal office at plosone@plos.org. Please include the following items when submitting your revised manuscript:A rebuttal letter that responds to each point raised by the academic editor and reviewer(s). You should upload this letter as a separate file labeled 'Response to Reviewers'.A marked-up copy of your manuscript that highlights changes made to the original version. You should upload this as a separate file labeled 'Revised Manuscript with Track Changes'.An unmarked version of your revised paper without tracked changes. You should upload this as a separate file labeled 'Manuscript'.

We look forward to receiving your revised manuscript.

Kind regards,

Claudia Trindade Mattos, Ph.D.

Academic Editor

PLOS ONE

Journal Requirements:

2. Please include a caption for figures 1, 2 and 3.

Additional Editor Comments:

We have carefully reviewed your work and found it to be both relevant to the field and methodologically sound. The study design demonstrates careful consideration, and the manuscript is generally well-written.

Based on the peer review process, we invite you to submit a revised version that addresses the concerns raised by the reviewers, which will help strengthen the study and make it adequate for publication.

Please provide a point-by-point response to the reviewers' comments along with your revised manuscript, clearly indicating where changes have been made in the text.

Reviewers' comments:

Reviewer's Responses to Questions

**Comments to the Author**

1. Is the manuscript technically sound, and do the data support the conclusions?

Reviewer #1: Yes

Reviewer #2: Partly

2. Has the statistical analysis been performed appropriately and rigorously? 

Reviewer #1: I Don't Know

Reviewer #2: Yes

3. Have the authors made all data underlying the findings in their manuscript fully available?

Reviewer #1: Yes

Reviewer #2: Yes

4. Is the manuscript presented in an intelligible fashion and written in standard English?

Reviewer #1: Yes

Reviewer #2: Yes

5. Review Comments to the Author

Reviewer #1: Thank you for the opportunity to read this interesting manuscript. The aim was to evaluate the clinical transfer accuracy of partially enclosed single hard vacuum-formed trays based on three-dimensional (3D) printed models for lingual bracket indirect bonding.

1) Abstract

“Transfer errors were statistically below the clinically acceptable thresholds for all dimensions but torque.”

It seems that this sentence is not complete.

2) In my opinion, this work goes against the current trend of indirect bonding techniques. Nowadays some software allow for building the tray virtually and direct printing it. It might be interesting to assess the accuracy of 3D printing trays for transferring the bracket position.

3) “For each tooth, the brackets on both the target and reference data were aligned with the bracket patch. The 89 bracket patch was then attached to the target data followed by superimposing the target data onto the 90 reference data using a local best-fit algorithm that considered only the tooth surfaces.”

I think that it is not clear if the region of reference for the superimposition was the bracket patch or the tooth surface.

4) Considering that the superimposition was based on the tooth surface (best-fit), how was standardized the cropping to obtain the same size and shape of the crown for both time points?

“The post-bonding scan data and the virtual model with ideal bracket positions, serving as the target and reference data, were segmented into individual teeth, removing the gingiva and retaining only the clinical crowns and brackets.”

5) “The manufacturer-provided virtual bracket patches without bracket bases were set at the origin of a coordinate system. For each tooth, the brackets on both the target and reference data were aligned with the bracket patch.”

“The new coordinates of the bracket patch would indicate the bracket transfer errors (Fig 2).”

I suggest that the authors add description about how the differences between the target and reference data were obtained, such as:

Which landmarks were used to take the linear measurements?

Which planes or lines were used to take the angular measurements?

Which software was used to obtain the measurements?

Reviewer #2: Dear Authors,

Congratulations on your manuscript. The paper is well written and well designed. Has several limitations and interpretation of the results should be done carefully as you described. I suggest publication.

6. PLOS authors have the option to publish the peer review history of their article (what does this mean?). If published, this will include your full peer review and any attached files.

Reviewer #1: No

Reviewer #2: No

---

## [Author Response · Author response to Decision Letter 0]

14 Nov 2024

November 14th, 2024

Dear Editorial Board, PLOS ONE,

We have revised the manuscript thoroughly according to the comments of the reviewers. Any revisions made in our manuscript document were highlighted in red. Please help us review the manuscript again.

Sincerely,

Reviewer #1: Thank you for the opportunity to read this interesting manuscript. The aim was to evaluate the clinical transfer accuracy of partially enclosed single hard vacuum-formed trays based on three-dimensional (3D) printed models for lingual bracket indirect bonding.

1) Abstract

“Transfer errors were statistically below the clinically acceptable thresholds for all dimensions but torque.”

It seems that this sentence is not complete.

Thank you for your comment, we have made this sentence more clear:

Transfer errors were statistically below the clinically acceptable thresholds for all dimensions except torque.

2) In my opinion, this work goes against the current trend of indirect bonding techniques. Nowadays some software allow for building the tray virtually and direct printing it. It might be interesting to assess the accuracy of 3D printing trays for transferring the bracket position.

Thank you for your comment. We have added the comparison with 3D-printed trays to the discussion section:

This study assesses a modified design for vacuum-formed trays, diverging from the current trend of utilizing directly 3D-printed trays for indirect bonding. However, using single vacuum-formed trays with 3D-printed models is more cost-effective than direct 3D-printing of trays, due to simplified design and reduced post-processing requirements [12]. As no studies in the literature have evaluated the accuracy of 3D-printed indirect bonding trays for lingual brackets, this study's findings are compared with those from a study using 3D-printed trays for labial brackets [4]. That study reported mean transfer errors of 0.10 mm mesiodistally, 0.10 mm buccolingually, 0.18 mm occlusogingivally, 2.47° in rotation, 2.01° in tip, and 2.55° in torque. While the linear transfer errors were within the clinically acceptable threshold, the angular errors were not. Notably, all transfer errors in that study exceeded those observed in the current study. This discrepancy may be attributed to the flexibility of the tray material and the inadequate bracket retention within the tray, as silicone and modeling wax were necessary to secure brackets in a similar study utilizing 3D-printed trays [9]. Furthermore, as both the 3D-printing material and the orthodontic adhesive are light-cured, their adhesive natures may interact, potentially hindering tray removal and causing tearing [12].

3) “For each tooth, the brackets on both the target and reference data were aligned with the bracket patch. The bracket patch was then attached to the target data followed by superimposing the target data onto the reference data using a local best-fit algorithm that considered only the tooth surfaces.”

I think that it is not clear if the region of reference for the superimposition was the bracket patch or the tooth surface.

Thank you for your comment. We have made the superimposition process more clear:

For each tooth, the brackets on both the target and reference data were initially aligned with the bracket patch, constituting the first and second superimpositions. Subsequently, the bracket patch was attached to the target data followed by superimposing the target data and bracket patch combination onto the reference data using a local best-fit algorithm that considered only the tooth surfaces. 

4) Considering that the superimposition was based on the tooth surface (best-fit), how was standardized the cropping to obtain the same size and shape of the crown for both time points?

“The post-bonding scan data and the virtual model with ideal bracket positions, serving as the target and reference data, were segmented into individual teeth, removing the gingiva and retaining only the clinical crowns and brackets.”

Thank you for your comment. We have added more descriptions on the 3D-data editing:

The post-bonding scan data and the virtual model with ideal bracket positions, serving as the target and reference data, were first roughly aligned. Then, they were segmented into individual teeth, removing the gingiva and retaining only the clinical crowns and brackets. This initial alignment ensured consistent cropping and facilitated the subsequent superimposition based on the tooth surface.

5) “The manufacturer-provided virtual bracket patches without bracket bases were set at the origin of a coordinate system. For each tooth, the brackets on both the target and reference data were aligned with the bracket patch.”

“The new coordinates of the bracket patch would indicate the bracket transfer errors (Fig 2).”

I suggest that the authors add description about how the differences between the target and reference data were obtained, such as:

Which landmarks were used to take the linear measurements?

Which planes or lines were used to take the angular measurements?

Which software was used to obtain the measurements?

Thank you for your comment. We have added more descriptions about the measuring process:

The centers of the manufacturer-provided virtual bracket patches’ without bracket bases were set at the origin of a coordinate system, with the x, y, and z-axis parallel to the mesiodistal edge, vertical edge, and buccolingual edge of the bracket slot, respectively.

The new coordinates of the bracket patch's center in the x, y, and z directions would indicate the bracket linear transfer errors. Meanwhile, the angular transfer errors would be represented by the rotation of the bracket slot's edges projected onto the respective coordinate planes (Fig 2). The 3D inspection software (Meshmixer, Autodesk, USA) was utilized to obtain these measurements.

Reviewer #2: Dear Authors,

Congratulations on your manuscript. The paper is well written and well designed. Has several limitations and interpretation of the results should be done carefully as you described. I suggest publication..

Thank you for your kind commendation.

---

## [Decision Letter · Decision Letter 1]

9 Dec 2024

Transfer accuracy of partially enclosed single hard vacuum-formed trays with 3D-printed models for lingual bracket indirect bonding: A prospective in-vivo study

PONE-D-24-35775R1

Dear Dr. Nguyen,

We’re pleased to inform you that your manuscript has been judged scientifically suitable for publication and will be formally accepted for publication once it meets all outstanding technical requirements.

Kind regards,

Claudia Trindade Mattos, Ph.D.

Academic Editor

PLOS ONE

Additional Editor Comments (optional):

Reviewers' comments:

Reviewer's Responses to Questions

**Comments to the Author**

1. If the authors have adequately addressed your comments raised in a previous round of review and you feel that this manuscript is now acceptable for publication, you may indicate that here to bypass the “Comments to the Author” section, enter your conflict of interest statement in the “Confidential to Editor” section, and submit your "Accept" recommendation.

Reviewer #1: All comments have been addressed

2. Is the manuscript technically sound, and do the data support the conclusions?

Reviewer #1: Yes

3. Has the statistical analysis been performed appropriately and rigorously? 

Reviewer #1: Yes

4. Have the authors made all data underlying the findings in their manuscript fully available?

Reviewer #1: Yes

5. Is the manuscript presented in an intelligible fashion and written in standard English?

Reviewer #1: Yes

6. Review Comments to the Author

Reviewer #1: Congratulations on your excellent work!

Thank you for addressing all my comments and suggestions properly.

7. PLOS authors have the option to publish the peer review history of their article (what does this mean?). If published, this will include your full peer review and any attached files.

Reviewer #1: No

---

## [Editor Report · Acceptance letter]

8 Jan 2025

PONE-D-24-35775R1 

PLOS ONE

Dear Dr. Nguyen, 

I'm pleased to inform you that your manuscript has been deemed suitable for publication in PLOS ONE. Congratulations! Your manuscript is now being handed over to our production team.

Kind regards, 

on behalf of

Dr. Claudia Trindade Mattos 

Academic Editor

PLOS ONE